# Bioaccessibility and Bioactivity of Cereal Polyphenols: A Review

**DOI:** 10.3390/foods10071595

**Published:** 2021-07-09

**Authors:** Borkwei Ed Nignpense, Nidhish Francis, Christopher Blanchard, Abishek Bommannan Santhakumar

**Affiliations:** 1School of Biomedical Sciences, Charles Sturt University, Locked Bag 588, Wagga Wagga, NSW 2678, Australia; bednignpense@csu.edu.au (B.E.N.); nfrancis@csu.edu.au (N.F.); cblanchard@csu.edu.au (C.B.); 2School of Animal and Veterinary Sciences, Charles Sturt University, Wagga Wagga, NSW 2650, Australia; 3Australian Research Council (ARC), Industrial Transformation Training Centre (ITTC) for Functional Grains, Graham Centre for Agricultural Innovation, Charles Sturt University, Wagga Wagga, NSW 2650, Australia

**Keywords:** cereal polyphenols, bioaccessibility, bioavailability, microbiota, intestinal barrier function, inflammation, oxidative stress

## Abstract

Cereal bioactive compounds, especially polyphenols, are known to possess a wide range of disease preventive properties that are attributed to their antioxidant and anti-inflammatory activity. However, due to their low plasma concentrations after oral intake, there is controversy regarding their therapeutic benefits in vivo. Within the gastrointestinal tract, some cereal polyphenols are absorbed in the small intestine, with the majority accumulating and metabolised by the colonic microbiota. Chemical and enzymatic processes occurring during gastrointestinal digestion modulate the bioactivity and bioaccessibility of phenolic compounds. The interactions between the cereal polyphenols and the intestinal epithelium allow the modulation of intestinal barrier function through antioxidant, anti-inflammatory activity and mucin production thereby improving intestinal health. The intestinal microbiota is believed to have a reciprocal interaction with polyphenols, wherein the microbiome produces bioactive and bioaccessible phenolic metabolites and the phenolic compound, in turn, modifies the microbiome composition favourably. Thus, the microbiome presents a key link between polyphenol consumption and the health benefits observed in metabolic conditions in numerous studies. This review will explore the therapeutic value of cereal polyphenols in conjunction with their bioaccessibility, impact on intestinal barrier function and interaction with the microbiome coupled with plasma anti-inflammatory effects.

## 1. Introduction

Cereals such as wheat, rice, oats, barley and sorghum are grown and consumed as staple diets globally [1]. Cereals are edible seeds or grains of the grass family, Gramineae that have structural similarities: an outer bran layer, endosperm and embryo [1]. Whole grains contain all of these three components whereas refined grains only have the endosperm [1]. Consumption of whole-grain cereals has been linked to a significant reduction in chronic diseases [2]. With the increase in chronic conditions such as obesity in developed countries and malnutrition in developing countries, cereal grains, due to their health benefits and affordability, provide nutritional value and food security, respectively. Substantial research has attributed the health benefits to bioactive compounds in whole grains, particularly micronutrients such as polyphenols.

Polyphenols are plant-based compounds (found in fruits, vegetables and grains) that have demonstrated bioactive properties including antithrombotic, anti-inflammatory and antioxidant properties both in vitro and in vivo [3,4,5,6]. They are a large group of heterogeneous compounds characterized by their basic structure consisting of a benzene ring with hydroxyl constituents [7]. For example, flavonoids, specific groups of polyphenols with A, B and C aromatic rings, exert antioxidant activity dependent on the functional groups attached and the degree of hydroxylation on the rings [8]. Among cereal grains, the phenolic composition may vary due to many factors such as environmental conditions, variety and pigmentation [9,10,11]. It is believed these polyphenols target pathways of inflammation through their antioxidant, metal-chelating and gene regulatory activities [12].

A study by Callcott et al. [13] demonstrated that consumption of pigmented rice alleviates plasma inflammatory markers in obese participants. The coloured rice, rich in anthocyanin—the polyphenol responsible for pigmentation—also displayed strong antioxidant and anti-inflammatory activity in vitro and ex vivo [14]. Unlike some other polyphenols, anthocyanins have a relatively higher bioaccessibility (the amount available for absorption) in the stomach and this may explain the systemic anti-inflammatory and antioxidant effects observed following acute consumption [15,16,17]. However, although the mode of action of polyphenols in vitro seem to correlate with in vivo findings, there is controversy surrounding bioavailability i.e., optimal physiological concentrations attainable in vivo that results in favourable biological activity (bioactivity) [18]. Following ingestion, gastrointestinal digestion and colonic bacterial metabolism are likely to significantly impact the bioaccessibility and bioavailability of the cereal polyphenols. For this reason, understanding the interaction between polyphenols and the intestinal milieu will help elucidate the positive health outcomes observed with consumption of polyphenol-rich cereals. Although studies in the past have attempted to investigate this interaction using polyphenols from various other food sources, an understanding of the role of cereal polyphenols is limited [19,20,21,22]. This review will provide an overview of the role of gastrointestinal (GI) digestion in the modulation of cereal polyphenol bioavailability. The impact cereal polyphenols may have on intestinal health and inflammation will be discussed by evaluating studies on dietary polyphenol bioaccessibility, the impact on intestinal barrier function, gut microbiome, and the associated antioxidant and anti-inflammatory plasma effects.

## 2. Cereal Polyphenols

The common phenolic compounds found in cereals include flavonoids and phenolic acids, classified as hydroxybenzoic or hydroxycinnamic acids [10]. Some phenolic compounds are rare and highly specific to a few cereal grains. Some examples include the 3-deoxyanthocyanin found in sorghum and avenanthramides found in oats [23]. The biological effects of these phenolic extracts can be attributed to their structure–activity relationships. Hydroxybenzoic and hydroxycinnamic acids possess free radical scavenging ability due to their polarity and hydroxyl groups (Table 1). One of the most abundant phenolic acids in cereals, ferulic acid, has one hydrogen atom from its hydroxyl group that can react with a free radical to exhibit antioxidant activity [10]. Other phenolic acids such as protocatechuic acid and caffeic acid exhibit antioxidant effects due to the resonance stabilization created by the presence of two hydroxyl groups in the ortho position [24,25,26]. However, depending on their electrochemical behaviour and oxidation potentials, Simić et al. [25] have shown that phenolic acids may act as either prooxidants or antioxidants. According to this study, the flavonols quercetin and rutin (quercetin-3-rutinoside), found in Buckwheat, demonstrate antioxidant activities which may be attributed to their structural features [27]. The antioxidant activity of quercetin and rutin is likely to be due to hydroxyls present at the C-3′, 4′ positions in the B ring [8].

Flavonoids contain functional groups that enhance both antioxidant and anti-inflammatory activity [28]. Flavones such as apigenin possess the 2,3-double bond and 4-keto group of the C ring that contribute to their antioxidant and anti-inflammatory activity and enhance affinity to target protein receptors [28]. Flavonoid glycosides such as cyanidin 3-glucosides exhibit antioxidant and anti-inflammatory activity and can be absorbed intact from the GI tract but with low bioavailability (1–100 nmol/L of total anthocyanin plasma concentration following doses of 0.7–10.9 mg/kg in human studies) [29,30]. On the other hand, the phenolic compound avenanthramide (AVN) is a conjugate of phenolic acid and an anthranilic acid and is known to have a relatively higher bioavailability (374.6 nmol/L of total AVN plasma concentration following 1 g avenanthramide-enriched mixture from oats) [31,32]. There are numerous AVN forms but AVN A, AVN B and AVN C are the main types (Table 1) Among the AVNs, AVN C exhibits the strongest antioxidant activity. AVN C requires a 0.074 μmoles concentration to achieve a 50% reduction in the free radical 2,2-diphenyl-1-picrylhy-drazyl compared to 0.198 μmoles and 0.105 μmoles concentration required by AVN A and AVN B, respectively [33]. This is likely to be due to the free radical scavenging of the 2-hydroxyl groups on the phenolic acid aromatic ring [31,34]. However, most of these structure–activity studies on cereal polyphenols have been undertaken in vitro and thus may not reflect the bioactivity observed in vivo especially after cereals undergo GI digestion.

### 2.1. GI Digestion of Cereal Polyphenols

#### 2.1.1. Oral and Gastric Digestion

Cereal grains when ingested travel first through the GI tract before being metabolised further by the liver and then entering the systemic circulation (Figure 1). During mastication in the oral cavity, chewing and salivary amylase break down the starch, mechanically and chemically, within the cereal matrix into a bolus [35]. The polyphenol-containing bolus then travels to the stomach where it undergoes gastric digestion involving gastric juice containing hydrochloric acid, pepsin, lipase, mucus, electrolytes and water [35]. Hydrochloric acid contributes to an acidic pH environment in the range of 1.3 to 2.5 that favours the denaturation of cereal peptides [36,37]. Some polymeric polyphenols such as procyanidins can be degraded into smaller units during gastric digestion, but there is limited information on the impact gastric digestion has on the bioactivity of cereal polyphenols [38]. Absorption of polyphenols in the stomach is limited yet some anthocyanin glycosides may be absorbed quickly in the gastric phase possibly through a bili-translocase, an organic anion membrane transporter, and appear in the bloodstream [16,39]. Evaluation of the structure–activity relationship of mono- and di-glucosyl anthocyanins reveals that they are better ligands for the bili-translocase than their corresponding aglycones [39].

#### 2.1.2. Small Intestinal Metabolism and Absorption

In the small intestines, the digesta from the stomach is neutralised by sodium hydroxide allowing intestinal enzymes to catabolize the food matrix [37]. Pancreatic juices containing bile salts, electrolytes and enzymes including pancreatin, proteases and lipases interact with the digesta. Bile salts allow for the micellization of lipophilic compounds [40]. However, it is postulated that, due to the hydrophilic nature of some glycosylated flavonols and derivatives of hydro-cinnamic acid, they may be readily soluble in the aqueous environment, whereas less soluble phenolic compounds such as flavonoid aglycones or procyanidins are bound to dietary fibre and proteins [35]. The stability of the phenolic compounds has been attributed mainly to the pH, with some phenolic compounds being degraded through non-enzymatic oxidisation. Anthocyanins, however, are relatively sensitive to pH in the intestinal phase. Anthocyanin recovery in the intestinal environment (neutral to basic pH) is lower than in a gastric environment (acidic pH) [41,42]. The discrepancy in the quantification of anthocyanins is due to the pH-dependent equilibrium of the flavylium cation form to other related structural forms above pH 2 [30]. Despite poor recovery after digestion, studies show that anthocyanins can be efficiently absorbed in jejunum tissue [43,44].

It is unclear what specific transport mechanisms are involved in polyphenol intestinal absorption. For monomeric compounds such as cinnamic acid and ferulic acid, a Na+ dependent transport mechanism has been postulated [45,46]. Flavonoid glycosides on the other hand can be transported in enterocytes by sodium-dependent glucose transporter [47,48]. Two enzymes, cytosolic β-glucosidase (CBG) and lactase phlorizin hydrolase (LPH) play a role in the pathway for de-glycosylation of flavonoids. LPH catalyses the hydrolysis of some glucosides on the brush border of the enterocytes (allowing aglycone form to be absorbed, possibly by diffusion) and CBG hydrolyses glucosides once absorbed into the enterocytes [49]. However, anthocyanins such as cyanidin 3-glucosides (C3G) are not substrates to either CBG or LPH but their absorption may be competitively inhibited by the presence of other flavonoids [30,50]. Nonetheless, the lack of enzymatic degradation of some polyphenols as well as their interaction with other dietary antioxidants in the cereal matrix may allow them to accumulate in the colon and exert bioactive functions therein [51].

#### 2.1.3. Colonic Metabolism

Polymeric or oligomeric compounds such as pro-anthocyanidins, due to their high molecular weight, are unlikely to be absorbed in the small intestine, but pass through to the colon [50]. Pro-anthocyanidins are not degraded by the acidic condition of the stomach and consequently, due to their poor absorption in the small intestine, may act as key players in cereal-mediated bioactivity in the colon [52]. In the colon, the microbial enzyme rhamnosidase hydrolyses polyphenols conjugated to a rhamnose sugar moiety to facilitate their absorption [46]. Dietary fibres such as xylose, cellulose, β-glucans, and arabinoxylans that are bound to polyphenols are substrates of microbial catabolism resulting in the production of short-chain fatty acids and the release of polyphenols for absorption [53]. Ferulic acid, found in the bran and aleurone layer of cereals, is usually bound to arabinoxylans and can be released by hydrolysis of ester linkages [54]. The absorption in the intestines is due to the size or number of sugar moieties such as arabinose and xylose.

#### 2.1.4. Extended Metabolism and Elimination

The pathway of metabolism for polyphenols is similar to xenobiotics (substances foreign to the body such as environmental chemicals and drugs) and thus involves extensive metabolism to counteract any potential toxic effects [18]. During the first pass metabolism of polyphenols, three main conjugations occur in the small intestine and the liver—glucuronidation, sulfation and methylation [55,56]. It is believed that the bioactivity may be reduced due to alteration in solubility and molecular weight of these bio-transformed phenolic compounds. Interestingly, sulphate and glucuronate metabolites of ferulic acid and caffeic acid have been shown to retain their strong bioactivity when compared to their parent compounds [57]. This highlights the possibility of cereal phenolic metabolites to retain strong bioactivity locally in the gut and systemically in plasma.

Following extensive metabolism of polyphenols, the half-life in plasma has been reported to be short [46]. However, it has been suggested that the resulting low plasma concentrations are an underestimation, as some phenolic compounds have an affinity to plasma proteins such as albumin [46,58]. This warrants confirmation by investigating polyphenolic binding affinity using improved detection and measurement tools. Examination of urinary and faecal excretion demonstrates that cereal consumption results in lower molecular weight phenolic compounds such as ferulic acid, dihydro-ferulic acid, hippuric acid, and hydroxy-hippuric acid [59,60]. These phenolic compounds may result from the absorption of compounds from the small intestine via colonic catabolism of polymeric compounds, or release compounds bound to the cereal fibre. The impact of the phenolic compounds on the intestinal environment is particularly important as the compounds metabolised in the liver return to the small intestines via the bile duct in a pathway known as enterohepatic recirculation resulting in more exposure to bio-transformed polyphenols [61]. Further investigation into the impact of this pathway on the biotransformation and bioactivity of polyphenols is warranted.

## 3. Polyphenol Bioaccessibility in the GI Tract

Bioaccessibility can be defined as the quantity of a compound available for absorption after GI digestion whereas bioavailability refers to the extent to which a compound enters the systemic circulation to exert bioactivity. The GI tract plays a key role in modulating the bioaccessibility and subsequent bioavailability of polyphenols in vivo. However, inter-individual variation creates challenges in analysing bioaccessibility and, thus, an in vitro GI digestion model provides a more standardised approach in analysis [62,63]. The model simulates the phases of GI digestion including the oral, gastric and intestinal phases. To determine the recovery of polyphenols post GI digestion, two common models used are small intestine bioaccessibility large intestine bioaccessibility. The former focuses on the amount and type of polyphenols that are absorbed and transported across the small intestine and other upper GI organs whereas the latter focuses on the polyphenols and metabolites absorbed and transported after microbial metabolism in the colon. With both approaches, dialysis tubes or Caco-2 cell monolayers are used to mimic the absorption or transport across the intestines [62].

Bioaccessibility studies on cereal polyphenols are limited, however, evaluating studies from other polyphenol-rich foods may provide insights into their potential bioaccessibility, and inform on the design of bioaccessibility experiments. In the study by Hilary, et al. [64] researchers investigated the phenolic characterisation and bioaccessibility of three different matrices of date seed (powder, bread and phenolic extract). Results indicated that date seed powder and extract recovered more phenolic compounds than the bread form. This may have been attributed to the stronger binding of phenolic compounds to dietary fibre in the bread form. These findings highlight the influence type of food matrix and processing may have on the bioaccessibility of polyphenols, especially those derived from cereals. Furthermore, it was observed that the phenolic compounds increased as digestion progressed (vanillic acid recorded 91% and 89% recovery following digestion of the powder and extract form). This observation agreed with the study by Chandrasekara and Shahidi [65] which demonstrated that total phenolic content increased during in vitro GI digestion of cooked millet grains. The antioxidant activity increased 20-fold in response to the increased phenolic content after digestion. However, the impact on polyphenol antioxidant potential post digestion may vary depending on the cereal matrix. Pigmented cereals rich in anthocyanins have not been thoroughly investigated but are likely to show significant reductions in their antioxidant activity following GI digestion. This is evidenced in the study by Bouayed, et al. [66] that demonstrated an increase in phenolic acids and flavonoids but a significant reduction in antioxidant activity after digestion of different apple varieties. These results may be due to poor recovery and loss of free radical scavenging functional groups of some of the polyphenols, especially anthocyanins, after the intestinal digestion phase.

Interestingly, anthocyanins and phenolic acids in purple rice phenolic extracts have been shown to exhibit antioxidant and anti-inflammatory effects in vitro [13]. However, the chemical and enzymatic conditions of GI digestion are likely to significantly impact their bioaccessibility and bioavailability. Similar to the study of Hilary et al. [64], coupling in vitro GI digestion and Caco-2 transport may reveal the predominant cereal phenolic compounds and anthocyanin metabolites that are bioaccessible. Following in vitro GI digestion of date seeds, Hilary et al. [64] demonstrated that, among the polyphenols recovered, phenolic acids were the predominant compounds transported across a Caco-2 cell monolayer. In comparison, a cross-over dietary intervention trial employing the same three forms of the same date seeds observed significant antioxidant effects in blood in conjunction with an abundant release of polyphenol metabolites including simple phenolic acids in the urine [67]. Together, these studies display the potential of bioaccessibility models to predict in vivo bioavailability of polyphenols. Furthermore, the study by Lila et al. [68] indicated that berry-derived anthocyanins which showed bioaccessibility in vitro were also observed to be bioavailable in vivo using a radiolabelling strategy. This approach will be valuable in evaluating bioavailability and bioactivity of pigmented cereals in the future. However, in the meantime more in vitro studies evaluating cereal polyphenols bioaccessibility are warranted.

One in vitro study employing Zimbabwean wild cereal grains showed that the bioaccessibility of phenolic compounds was higher in the colon than in the small intestines [69]. The cereal grains with higher fibre content displayed lower bioaccessibility levels of polyphenols indicating the ineffectiveness of digestive enzymes in releasing conjugated polyphenols when compared to microbial enzymes (Table 2). Nevertheless, food processing techniques including extrusion, malting, and fermentation treatment can increase the bioaccessibility of phenolic compounds, particularly those with strong bioactive properties such as ferulic acids and AVNs (Table 2) [70,71]. It should be noted that there is a lack of a standardised approach to measuring bioaccessibility and predicting polyphenol bioavailability. Bioaccessibility studies employ a range of models from simple static models involving only chemical and enzymatic biotransformation to dynamic models from The Netherlands Organisation for applied scientific research (TNO). The TNO gastrointestinal model (TIM-1) simulates peristaltic movement and can closely mimic conditions in both the upper and lower intestines [62]. However, the bioaccessibility of polyphenols post-digestion can be reduced significantly, and consequently in vivo bioactivity may be attributed to the metabolites present. Conversely, a study demonstrated that blackberry extract still retained its protective effect against oxidative damage [72]. These findings are likely attributed to the strong antioxidant phenolic compounds C3G and ellagic acid recovered after digestion. Since cereal polyphenols travel to the colon during digestion, it will be of interest to investigate the impact of simulated GI digestion on the bioactive properties of the polyphenols and their potential impact on gut health.

## 4. Impact of Polyphenols on Intestinal Barrier Function

Without any pathological impairment, the intestinal tract, with the aid of immune cells, mucus and intact epithelium, prevents oxidative damage and the penetration of harmful toxins and pro-inflammatory mediators. On the other hand, it allows the selective permeability of essential nutrients from the lumen. The latter is known as intestinal permeability and is dependent on the integrity of the epithelial barrier [77]. Intestinal diseases and metabolic conditions such as type 2 diabetes and obesity have been linked to impairments of the intestinal integrity and consequently the function of the intestinal barrier [77].

Several studies have focused on understanding how individual phenolic compounds and some dietary extracts impact intestinal barrier function, but there is limited information on cereal phenolic extracts [22,78,79,80]. Intestinal barrier function can be conserved by the alleviation of oxidative stress and inflammation, induction of mucus production and reduction of intestinal permeability. In vitro cell culture models are often used to simulate oxidative stress- and inflammation-induced damage in the intestines through the use of oxidants (such as hydrogen peroxide and oxysterols, cholesterol auto-oxidation products) and pro-inflammatory mediators such as interleukin 1 beta (IL-1β), tumour necrosis factor-α (TNF-α) or lipopolysaccharides (LPS), respectively [22,81]. Upon exposure to polyphenols, the protective response of the intestines may include the secretion of mucus (mucin proteins secreted by goblet cells to provide a physical barrier) and reduced intestinal permeability indicated by the expression of tight junction proteins (Figure 2) [80,81]. Tight junction proteins such as occludin, zona occludens and other adhesion complexes seal adjacent cells to regulate permeability and maintain barrier integrity [82]. Their expression and distribution in polarized intestinal cell monolayers are measured as trans-epithelial electrical resistance (TEER) [82]. On a molecular level, the impact of polyphenols on specific signalling pathways in enterocytes can be investigated by measuring the expression or activity of antioxidant enzymes such as glutathione and superoxide dismutase, and antioxidant and anti-inflammatory regulatory genes such as nuclear factor erythroid 2-related factor 2 (NrF2) or nuclear factor kappa-light-chain-enhancer of activated B cells (NF-kB), respectively [83].

Interestingly, catechins commonly found in tea and some cereals have the potential to impact the signalling pathways in enterocytes. A study employing catechin-rich Fuzhuan brick-tea extract demonstrated that the extract was able to inhibit hydrogen peroxide-induced oxidative damage through the release of antioxidant enzymes and the inhibition of lipid peroxidation in the enterocytes [78]. A recent study showed that catechin-rich green tea could mitigate gliadin-induced inflammation and intestinal permeability [79]. In comparison with the previous study, the catechins attenuated inflammation potentially by complexing with gliadin to prevent the formation of gliadin peptides that mediate intestinal inflammation. Interestingly, pigmented sorghum has been characterised showing catechins to be the most abundant phenolic compound (2.11 ± 0.47 mg 100 g^−1^ gallic acid equivalents in black sorghum) with a relatively high free radical scavenging activity (5.36 ± 0.66 mg 100 g^−1^ Trolox equivalents in brown sorghum) [9]. However, despite knowledge of their antioxidant potential, there are limited studies of the impact of catechin-rich cereals on enterocyte signaling pathways. Moreover, individual phenolic compounds such as epicatechin gallate and quercetin have shown the ability to modulate mucin (MUC2, MUC3, MUC13 and MUC17) expression and secretion from intestinal goblet cells [80]. These bioactive effects were observed at physiologically relevant concentrations, but the underlying mechanisms have not been elucidated.

It is believed that polyphenols interact with multiple aspects of cell signalling to attenuate inflammation and oxidative stress and to induce mucus production [19]. Flavonoids may inhibit the induction of signalling cascade by scavenging free radicals or interfering with the binding of inflammatory mediators to receptors. When absorbed into the enterocytes, the phenolic compounds may interfere with the mitogen-activated protein kinase (MAPK) pathway—a complex array of intracellular signalling cascades involved in cell maturation, cell death and, more importantly, intestinal barrier function. Polyphenols such as catechin may improve barrier function by attenuating the MAPK signalling cascade. This involves the inhibition of c-Jun amino-terminal kinase (JNK), p38 and extracellular signal-regulated kinase (ERK1/2) phosphorylation that occur in NF-kB pro-inflammatory signalling [84]. In addition to MAPK signalling, polyphenols attenuate NF-kB activation by directly inhibiting I kappa B-alpha phosphorylation [85]. On the other hand, in the presence of oxidative stress, phosphorylation of ERK1/2 activates Nrf2 leading to antioxidant protein expression. Polyphenols can activate the Nrf2 gene by enhancing the phosphorylation of ERK1/2, phosphoinositide 3-kinase/Protein kinase B (PI3K/Akt) pathway or inhibiting Keap1 from inactivating Nrf2 through its binding [84]. Interestingly in the presence of phenolic compounds, ERK1/2, PI3K/Akt and p38 are activated, thus resulting in the tight junction protein expression to enhance barrier integrity. In the goblet cells, mucin gene expression has been shown to be stimulated by quercetin, enhancing the phosphorylation of ERK1/2, PKC and PLC [86]. Together these studies highlight the fact that polyphenols may improve barrier function by interacting with a common signalling cascade—the MAPK pathway. However, the modulation of these signalling may differ depending on the type of polyphenols present. A previous study indicating that at 50 μM the phenolic compounds chrysin and ellagic inhibited NF-kB activity in intestinal cells, whereas genistein and resveratrol increased it [85]. Thus, it is likely that modulation of MAPK signalling may differ in response to the phenolic composition of different food matrices.

With regards to cereals, there is limited information on the impact of the phenolic extracts on barrier function (Table 3). However, an in vivo study conducted in mice demonstrated that rice bran phenolic extract alleviated endotoxin-induced intestinal barrier dysfunction improving the expression of tight junction proteins [87]. These findings agree with an in vitro study that demonstrated that ferulic acid, a common bioavailable phenolic acid in rice, ameliorates LPS-induced barrier dysfunction by improving tight junction protein expression and activating the PI3K/AKT signalling pathway. Juxtaposed to each other, the studies demonstrate the potential effect of rice polyphenols on barrier integrity. Furthermore, pigmented cereals are rich in phenolic compounds capable of promoting barrier function. Animal studies employing a colitis mice model have demonstrated that supplementation with maize cultivars rich in flavano-4-ols and anthocyanin phenolic compounds reduced gut permeability and proinflammatory cytokine secretion, and enhanced mucus secretion [21,88]. The molecular mechanisms involved in the modulation of barrier function, however, need elucidating. Furthermore, since these studies detected changes in the microbiome, it will be interesting to investigate the potential impact of cereal phenolic compounds on the microbiome.

## 5. Cereal Polyphenol Impact on the Gut Microbiome and Plasma Anti-Inflammatory Status

Gut microbiota is a regulator of intestinal and systemic health that is gaining much attention due to its multifaceted role in modulating host immune response and metabolism [91,92]. Within the GI tract, an abundant and diverse microbial community (more than 300 trillion microbes) exists, with the colon being the most densely populated region [93]. Taxonomically, phyla present in the colon include Firmicutes, Bacteroidetes, Actinobacteria, Proteobacteria, Fusobacteria, and Verrucomicrobia, with Bacteriodetes and Firmicutes being the most abundant phyla [94,95]. These bacterial communities maintain intestinal barrier integrity by detoxifying xenobiotics, competing with pathogenic bacteria and keeping the mucosal immune system active [96,97]. With the consumption of cereals, the microbiota is predominantly involved in the fermentation of fibre to short-chain fatty acids (SCFAs) such as propionate, butyrate and acetate which in turn serve as an energy source for colonocytes and exert health-promoting effects on the intestinal epithelium [96]. The composition of the microbiome can be altered by factors including age, delivery pattern, drugs and diet [98]. Obesity induced by high-fat diet can lead to dysbiosis wherein there is an increase in the Firmicutes to Bacteriodetes ratio [99]. Metformin, a drug that regulates systemic metabolism by reducing blood glucose, has been shown to alter the gut microbiota composition, thereby suggesting the gut microbiome as a possible signalling pathway for the drug to exerts its physiological functions [100].

There is growing interest in understanding polyphenol interactions with gut microbiota as a key link to explain the health benefits observed in metabolic diseases [95]. Polyphenols are believed to demonstrate a reciprocal relationship with the gut microbiome—where on the one hand polyphenols are broken down into more bioactive metabolites, and on the other, there is a change in microbiome composition (prebiotic effect) all leading to favourable systemic effects [94,95]. In vitro fermentation studies and dietary intervention, trials have shown that whole grains, when compared to refined grains, enhance the growth of health-promoting bacteria such as *Akkermansia*, *Bifidobacterium* and *Lactobacillus* [94,95]. However, the mechanisms behind the interaction between whole-grain polyphenols and the microbiota in these studies are not well defined. There is some evidence that it is the bran constituents and not the polyphenols alone that alter the microbiota. Kristek, et al. [101] using an in vitro fermentation model of the gut microbiota showed that the whole oat bran, rather than its main bioactive polyphenols, induced an increase in *Bifidobacterium* and short-chain fatty acids. However, Wang et al. [34] demonstrated that the AVN C in oats are bio-transformed into metabolites such as caffeic acid, 5-hydroxyanthranilic and ferulic acid possibly through the cleavage of C7′-C8′ double bond and the cleavage of its amide bond (Table 1). These metabolites coupled with the parent compound AVN were shown in the study to induce apoptosis in colon cancer cells thus potentially promoting gut health. This presents an interesting area of research into clarifying mechanisms by which oat bran polyphenols mediate systemic effects by interacting with the gut microbiome in vivo. Moreover, other phenolic-rich cereal brans have exhibited favourable prebiotic effects.

An in vitro fermentation by Pham, et al. [102] demonstrated that red rice bran polyphenols increased the relative abundance of *Faecalibacterium—*beneficial bacteria producing butyrate. In the study, however, combined supplementation of ferulyated arabinoxylan oligosaccharides and rice bran phenolic extracts exerted a stronger prebiotic effect—that is, an increased abundance of butyrogenic bacteria, namely *Coproccus* and *Roseburi*. A similar synergistic effect was observed within the in vitro fermentation study of Ashley et al. [40]. Results demonstrated that Sumac and Black sorghum bran polyphenols independently stimulated the growth of *Roseburia* and *Prevotella* but worked together with a prebiotic, fructo-oligosaccharide to promote the growth of *Bifidobacterium* and *Lactobacillus*. Comparing the two studies, it is interesting to note that both sorghum and rice bran phenolic extracts are rich in pro-anthocyanidins. Pro-anthocyanidins have previously been reported to enhance the growth of beneficial bacteria such as *Bifidobacteria* and *Lactobacillus* and could play a key role in the prebiotic effects observed in cereal brans [103,104]. Furthermore, pro-anthocyanidins are converted by the gut microbiota into lower molecular weight phenolic acids which may have bioactive relevance in vivo [105]. However, the aforementioned batch fermentation studies did not focus on the phenolic metabolites produced, and thus it may be of interest to investigate the reciprocal interaction between pro-anthocyanidin-rich cereal and the gut microbiome. Moreover, although in vitro batch fermentation models offer a cost-effective approach to study interactions with the microbiome, they are a poor simulation of an in vivo state.

Human dietary intervention trials have been employed to investigate the reciprocal interaction of cereal polyphenols and the microbiome. Some trials have indicated a reduction in systemic inflammation but no significant change on gut microbiota when consuming whole grains as opposed to refined grains [61,99,106]. However, in these studies, the phenolic metabolites were not profiled in either plasma or excreta, thus making it difficult to ascertain potential mechanisms for polyphenol interactions with the microbiota. The choice of overweight and obese participants in intervention trials is appropriate as obesity is a chronic subclinical inflammation state associated with gut microbiome dysbiosis and metabolic syndrome risk factors such as dyslipidaemia and hyperglycaemia [95]. Interestingly, studies by Roager, et al. [107] and Kopf, et al. [108] demonstrated that after obese and overweight participants consumed whole grain there was a significant reduction in serum inflammatory markers but no significant change in the gut microbiome. In contrast, Vanegas, et al. [109] and Vitaglione et al. [60] observed a beneficial impact on the gut microbiome, wherein whole grain consumption significantly increased beneficial bacteria (*Bacteroides* and *Lactobacillus*) and decreased inflammatory bacteria *Enterobacteriaceae*. The inconsistent observations on gut microbiome in these studies may be due to other confounding factors such as inter-individual variations in participant race, age and gender. However, Vitaglione et al. [60] measured plasma phenolic metabolites to evaluate microbial metabolism of cereal polyphenols. Results indicated significant increases in serum phenolic metabolites after whole grain consumption which was associated with a significant reduction in TNF-α and IL-10. Reduction in TNF-α levels correlated with increased abundance of beneficial bacteria, thus highlighting a plausible association between microbiome modulation and plasma inflammatory status that needs further investigation. Nevertheless, when comparing the consumption of cereal grains with different phenolic profiles, there is evidence of varying impact on plasma inflammatory status.

Pigmented whole grain varieties have shown significant anti-inflammatory effects that can be attributed to their anthocyanin content. The study by Gamel, et al. [110] showed a significant decrease in TNF-α and an increase in glutathione (GSH), a marker antioxidant status in plasma after consumption of purple wheat. However, only modest differences were observed between the inflammatory status of participants consuming purple wheat and regular wheat. The lack of a strong difference between test groups may be because the total phenolic content of the grains did not vary significantly. Nevertheless, C3G-rich black rice has demonstrated therapeutic benefits by reducing C reactive protein (CRP) in coronary heart disease patients when consumed long term [111]. Purple rice and red rice have also demonstrated anti-inflammatory effects (reduction in IL-6, IL-10 and IL-12) and reduction in lipid peroxidation when consumed acutely in an obese cohort (Table 4) [13]. Anthocyanins in the purple cereals may contribute significantly to the plasma effects seen, but due to their low bioavailability, their respective metabolites in plasma may be the key bioactive players. Thus, further investigation is needed to identify phenolic metabolites produced after GI digestion and breakdown and their correlation with inflammatory changes observed.

## 6. Limitations

Polyphenols are not the only cereal constituents with health-promoting effects. Fibre, vitamins, and minerals are among the other ingredients that exert beneficial biological effects in vivo. The phenolic profile consisting of the total phenolic content and type of polyphenols within a cereal matrix may result in differential effects between and within cereal genotypes. Furthermore, the allergic potential of gluten-containing cereals also needs consideration as this may nullify the anti-inflammatory effects of phenolic compounds. Further clarification of the polyphenol–gluten interaction within a cereal matrix may lead to the design of less allergenic cereal products. Where allergenicity is not a concern, understanding the interaction between cereal phenolic compounds, the gastrointestinal environment and the intestinal barrier will help to elucidate the molecular signalling pathways underlining the anti-inflammatory and antioxidant responses exerted in vivo. The potential induction of mucin generation is also important and results in a protective barrier for the intestinal epithelium. However, mucin polymers form a barrier between cereal phenolic compounds in the lumen and the intestinal epithelium. Smaller molecular weight phenolic compounds may travel across the mucin marrier to interact with intestinal epithelium. Future studies are warranted to investigate this interaction in vivo. Interestingly, the gut microbiota plays an essential role in modulating specific phenolic compounds that come into contact with the intestinal epithelium. Furthermore, increases in mucin-degrading microbiota such as *Akkermansia* have been correlated with polyphenol intake in high fat diet-induced obese mice [99]. This highlights the role the interaction between microbiome, mucin and polyphenols can play in maintaining intestinal integrity. Moreover, the relationship between changes in microbiome and inflammation following cereal polyphenol intake needs investigation. The dose and duration of phenolic-rich cereals have to be considered as they may influence the physiological effects observed (Table 3 and Table 4). There is likely to be a complex mechanism in action, due to the rapid absorption of free phenolic compounds and the accumulation of fibre bound phenolic compounds in the colon. Phenolic metabolites and short-chain fatty acids from microbial metabolism, once absorbed, may contribute to the overall plasma anti-inflammatory and antioxidant status. However, well-controlled intervention trials will be needed to support this hypothesis.

## 7. Conclusions

Cereal polyphenols have the potential to modulate intestinal health and systemic inflammatory status. Polyphenol bioaccessibility is low but dependent on the type of polyphenol or cereal matrix involved. Phenolic compounds target specific mechanistic pathways involved in maintaining intestinal barrier function. When consumed, cereal polyphenols can potentially change the gut microbiome favourably and, in turn, improve plasma antioxidant and anti-inflammatory status. Nevertheless, there is limited data on the bioaccessibility and bioavailability of cereal polyphenols and their interaction with the intestinal barrier, gut microbiome, and plasma inflammatory mediators. Further studies in these areas may reveal a novel mechanism involved in polyphenol bioactivity and lead to the breeding and selection of polyphenol-rich cereal grains.

## Figures and Tables

**Figure 1 foods-10-01595-f001:**
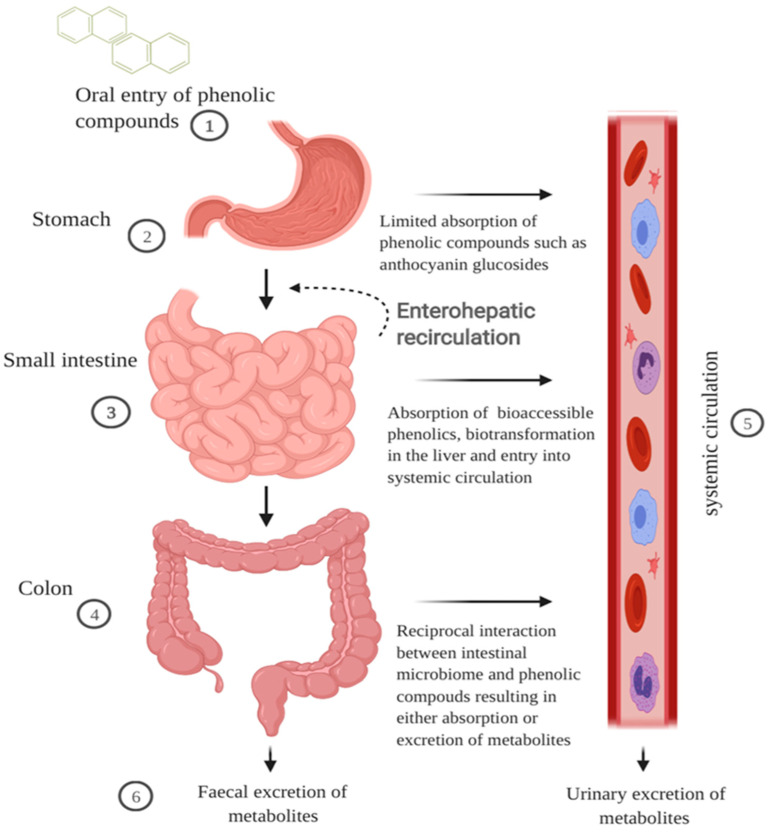
Schematic illustration of the impact of GI digestion in modulating bioavailability of cereal polyphenols. Cereal polyphenols travel through the digestive tract post-ingestion and the majority accumulate in the gut. (1) The cereal matrix forms a bolus and enhances the solubility of hydrophobic polyphenols. (2) Phenolic compounds such as anthocyanin glycosides can be absorbed quickly at the gastric phase. (3) In the small intestine, low molecular weight phenolic compounds are absorbed through active transport or sodium-dependent glucose transport. The first pass metabolism of polyphenols begins in the enterocytes and liver phenolic metabolites return to the intestines via the bile duct excretion. (4) The majority of polyphenols including polymeric compounds and phenolic acids bound to fibre are metabolised in the colon into smaller molecular weight forms that are bioaccessible. (5) Phenolic metabolites in the plasma are transported either freely or bound to blood proteins such as albumin to exert bioactivity in surrounding cells or tissues. (6) Phenolic metabolites are excreted in faeces or urine via the gut or blood, respectively. The figure was created with BioRender.com.

**Figure 2 foods-10-01595-f002:**
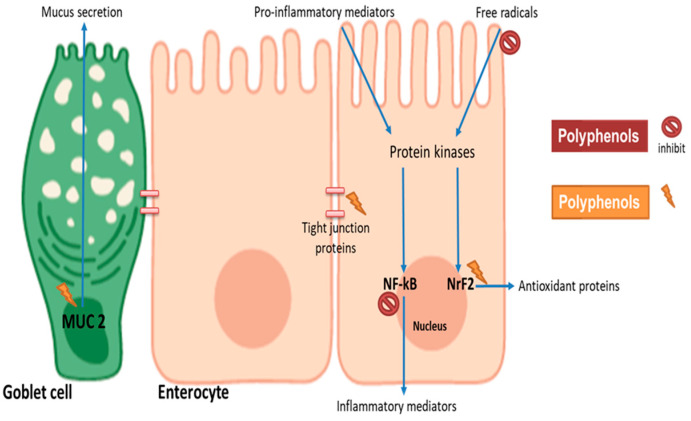
Potential mechanisms involved in polyphenol modulation of intestinal barrier function. Cereal polyphenols may attenuate pro-inflammatory and oxidative stress-induced barrier dysfunction by inhibiting NF-kB and stimulating Nrf2, respectively. Polyphenols may induce the expression of tight junction proteins to enhance barrier integrity. Mucin gene (MUC2) upregulation in goblet cells provides physical layer protection. The figure was created with BioRender.com.

**Table 1 foods-10-01595-t001:** Basic structures of common phenolic compounds found in cereals.

***Hydroxybenzoic acid***	***Hydroxycinnamic acid***
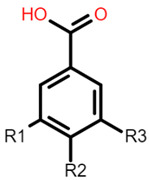 Gallic acid: R1 = OH; R2 = OH; R3 = OH Protocatechuic acid: R1 = OH; R2 = OH; R3 = H Vanillic acid: R1 = OH; R2 = OCH_3_; R3 = H Hydrobenzoic acid: R1 = H; R2 = OH; R3 = H	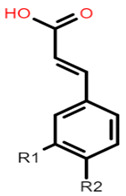 Caffeic acid: R1 = OH; R2 = OH Ferulic acid: R1 = OCH_3_; R2 = OH p-coumaric acid: R1 = H; R2 = OH
***Flavonoid backbone***	***Flavonol***
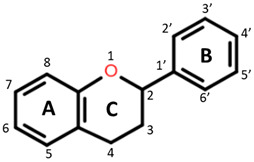 Flavanones: C4 keto group Flavanols: C4 keto group, C3 hydroxyl group Flavones: C2–C3 double bond, C4 keto group Flavan-3-ols: C3-hydroxyl group	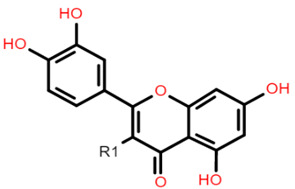 Quercetin: R1 = OH Rutin (quercetin 3-O glucoside): R1 = O-β-D-rutinoside
***Anthocyanin***	***3-Deoxyanthocyanidin***
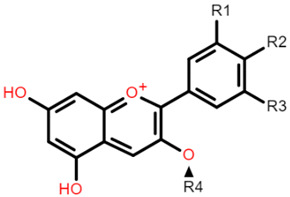 Cyanidin 3-glucoside: R1 = OH; R2 = OH; R3 = H; R4 = O-β-D-glucoside Delphinidin 3-glucoside: R1 = OH; R2 = OH; R3 = OH; R4 = O-β-D-glucoside Peonidin 3-glucoside: R1 = OCH_3_; R2 = OH; R3 = H; R4 = O-β-D-glucoside Petunidin 3-glucoside: R1 = OH; R2 = OH; R3 = OCH_3_; R4 = O-β-D-glucoside Malvidin 3-glucoside: R1 = OCH_3_; R2 = OH; R3 = OCH_3_; R4 = O-β-D-glucoside Pelargonidin 3-glucoside: R1 = H; R2 = OH; R3 = H; R4 = O-β-D-glucoside	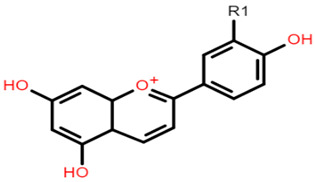 Apigeninidin: R1 = H Luteolinidin: R1 = OH
***Avenanthramide***	***Tannin repeating unit***
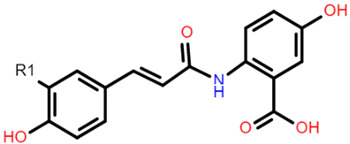 AVN A: R1 = H AVN B: R1 = OCH_3_ AVN C: R1 = OH	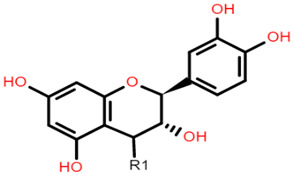 Catechin (or epicatechin) basic repeating unit: R1 = H Proanthocyanidin (or Procyanidin): R1 = Catechin

**Table 2 foods-10-01595-t002:** Impact of GI digestion on dietary polyphenol bioaccessibility.

Food Source	Study Design	Bioaccessibility Model	Findings	Ref.
**Four Luxembourgish apple varieties**	In vitro	Gastric and intestinal phase digestion combined with dialysis to assess recovery	- More phenolic acids and flavonoids are released during the gastric phase (65%) with a further release during intestinal digestion (<10%). - Anthocyanins not detected following intestinal digestion. - Antioxidant capacity of dialyzable phenolic acids were 57% and 46% lower than that of total antioxidants in fresh apples	[66]
**Grape seed phenolic extracts**	In vitro	Gastric, intestinal digestion, and in vitro ileal and faecal fermentation combined with a Caco-2 cell transport	- Monomeric phenolic compounds were significantly increased (*p* < 0.05) after gastric digestion. - Only microbial metabolites of the polyphenols were transported.	[73]
**Phoenix dactylifera L. (date) seeds; Date seed powder (DSP), date seed pita bread (DSB), date seed extract (DSE)**	In vitro	Gastric and intestinal phase digestion combined with Caco-2 cell transport	- Substantial phenolic acid recovery in DSE and DSP with vanillic acid recording 91% and 89% recovery respectively following in vitro digestion. DSB did not show any vanillic acid recovery. - Transport of DSE and DSP polyphenols including protocatechuic acid, p-hydroxybenzoic acid, caffeoylshikimic acid, p-coumaric acid, syringic acid hexoside and diosmin from across the cell monolayer. Whereas only protocatechuic acid and p-hydroxybenzoic acid were transported from DSB.	[64]
**Zimbabwean cereal grains**	In vitro	Gastric, intestinal digestion, and colonic fermentation combined with dialysis	- Higher bioaccessibility after colonic fermentation (62%) than GI enzymatic digestion (28%). - Eleusine corocana and red Sorghum bicolor had the highest polyphenol bioaccessibility; 31.57% and 30.64% respectively.	[69]
**Wheat, brown rice, oat grains**	In vitro	Gastric and intestinal digestion combined with dialysis	- The amount of bioaccessibile phenolic compounds in brown rice, oat and wheat was 528.99, 308.83, and 443.44 μg FAE/g DW respectively - After IECT, the total antioxidant activity of free form polyphenols in brown rice, wheat and oats was 63.06%, 36.55%, 14.39% respectively.	[74]
**Cookies from malted oat flours (avenanthramides)**	In vitro	Single bioreactor (oral, gastric and intestinal phases of digestion)	- Inclusion of 27% malted oat flour increased AVN content up to 10-fold. Likewise, bioaccessibility with a recovery of 1703 μg AVNs per 50 g of cookies from malted flour compared to 135 μg of control cookies (made from non-malted flour).	[71]
**Wheat bran**	In vitro	TIM-1 (upper GI digestion) and TIM-2 (human colon fermentation)	- Wheat bran ferulic acid bioaccessibility increased from 1.1% to 5.5% with enzymatic and fermentation treatment	[75]
**Extruded Barley and Oats**	In vivo (pigs)	N/A—blood and faecal samples collected at the start and during the trial	- Extrusion of the barley and oat grains increased the bioaccessibility of bound phenolic acid by 29% and 14%.	[70]
**Whole grain oat products**	In vitro	Gastrointestinal digestion (oral, gastric and intestinal) coupled with Caco-2 cell uptake	- Puffed oat cereal had a higher bioaccessibility of AVN (89%) compared with matching wet cook porridge (19.1%) from the same flour. Intestinal uptake for the digesta was low (between 0.16% and 2.71%) for all oat products.	[76]
**Cooked millet grains (kodo, finger, proso, foxtail and pearl)**	In vitro	Gastric, intestinal digestion, and colonic fermentation combined with dialysis	- Total phenolic content ranged from 10.2 to 26.9 μmol FAE/g DW at the end of the gastric phase and this was 2–5 times more than that released from aqueous extraction. The completion of GI digestion significantly increased total phenolic content to a range from 12.7 to 35.4 μmol FAE/g DW. After in vitro colonic fermentation the phenolic content was 4.5 to 17.1 μmol FAE/g DW. - The antioxidant activity was significantly increased after digestion of the millet grains (more than 20-fold increase when measured using an ABTS radical anion assay).	[65]

N/A, Not Applicable; AVN, avenanthramide; IECT, Improved Extrusion Cooking Treatment; TIM, TNO gastrointestinal model; DSE, Date Seed Extract; DSP, Date Seed Powder; DSB, Date Seed Bread; DW, Dry Weight; FAE, Ferulic Acid Equivalent; ABTS, 2, 2′-Azino-Bis-3-Ethylbenzothiazoline-6-Sulfonic Acid.

**Table 3 foods-10-01595-t003:** The impact of dietary polyphenols on intestinal barrier function compromised by inflammation and oxidative stress agonists.

Polyphenol/Food Source	Significant Dose	Study Design	Agonist	Cell Culture Model	Biomarkers	Findings	Ref.
Cyanidin-3-glucoside (C3G)	20 μM	In vitro	TNF-α	Caco-2/HUVEC/ leucocytes	E-selectin, VCAM-1, NF-κB and TNF-α and IL-8	- Reduction in nuclear translocation of NF-κB and TNF-α and IL-8 gene expression in Caco-2 cells. - Inhibition of TNF-α-stimulated Caco-2 cells endothelial cell activation as indicated by increased E-selectin and VCAM-1 mRNA.	[89]
Chlorogenic acid and epicatechin gallate	10 μM	In vitro	N/A	Caco-2 and HT29-MTX (mucus-secreting goblet cells)	Mucin protein and gene expression	- No significant difference in MUC2 protein levels post-treatment. - Significant downregulation and upregulation of MUC2 and MUC17 genes respectively.	[80]
Ferulic acid	100 μM	In vitro	LPS	Caco-2 cells	microRNA expression, epithelial permeability, TJ proteins., P13K and AKT protein	- Inhibition of LPS-induced decrease in TJ protein expression and microRNA expression. - Activation of P13K/AKT signalling in the presence of LPS induced barrier dysfunction	[90]
Polyphenol-rich propolis extract (PPE)	50 µg/mL	In vivo & in vitro (18 male Sprague-Dawley rats; 300–320 g)	N/A	Caco-2 cells	TEER, LY flux, TJ proteins (Zonulin and occludin tight junction protein gene expression), AMPK and ERK expression	- Increased TEER and decreased lucifer yellow flux. - Increased expression of the tight junction (TJ) loci occludin and zona occludens (ZO)-1. - AMPK, ERK1/2, p38, and Akt signalling was activated in response to PPE.	[20]
Sardinian wine extracts (red Cannonau and white Vermentino)	25 μg/mL	In vitro	Oxysterol	Caco-2 cells	NOX1, IL6 and IL-8	- Red Cannonau wine attenuated IL-6 and IL-8 expression but white Vermentino wine did not. - The activation of NOX1 prevented by both extracts.	[22]
Fuzhuan brick-tea extract	25, 100 and 200 μg/mL	In vitro	Hydrogen peroxide	Caco-2 cells	GSH, IL-8 and MDA	- Increased level of GSH but reduction in lipid peroxidation (reduced MDA levels) and IL-8 secretion.	[78]
Green tea polyphenols	1 mg/mL	In vitro	Gliadin	Caco-2 cells	Intestinal permeability, IL-6 and IL-8	- Reduction in gliadin-stimulated monolayer permeability and IL-6 and IL-8 release.	[79]
Flavan-4-ols enriched maize	15% or 25% (*w*/*w*)	In vivo (C57BL6 mice for 6 weeks)	Carboxymethylcellulose (CMC)	N/A	IL-6 and intestinal permeability	- Alleviated the CMC- induced increase in IL-6, intestinal permeability and enhanced mucus thickness.	[88]
Flavan-4-ols- and anthocyanin- enriched maize	25% (*w*/*w*)	In vivo (C57BL6 mice)	Dextran sulfate sodium (DSS)	N/A	Intestinal permeability, IL-1β and IL-6	- Attenuation of DSS-induced colitis by reducing intestinal permeability and proinflammatory secretion.	[21]
Rice bran phenolic extract (RBPE)	100 and 200 mg/kg	In vivo (C57BL6 mice)	Ethanol	N/A	TJ proteins (ZO-1, Claudin), TNF-α, IL-1β, IL-6, IFN-γ, and MCP-1	- Increased TJ protein expression and reduced serum proinflammatory mediators.	[87]

N/A, Not Applicable; C3G, cyanidin 3-glucoside; DSS, Dextran sulfate sodium; RBPE, Rice bran phenolic extract; CMC, Carboxymethylcellulose; HUVEC, Human umbilical vein endothelial cells; PPE, Polyphenol-rich propolis extract; VCAM-1, Vascular Cell Adhesion Molecule 1; TJ protein, Tight junction protein; TEER, transepithelial electrical resistance; MDA, malondialdehyde; IL, interleukin; IFN, interferon; MCP-1, monocyte chemoattractant protein-1; NOX1, NADPH oxidase 1; AMPK, adenosine monophosphate activated-protein kinase; ERK, extracellular signal-regulated kinase; P13K, phosphoinositide 3-kinases; AKT, Protein kinase B; ZO-1, Zonula occludens-1; LY flux, Lucifer yellow flux; GSH, Glutathione; LPS, Liposaccharide; NF-κB, nuclear factor kappa-light-chain-enhancer of activated B cells.

**Table 4 foods-10-01595-t004:** Impact of cereal grain polyphenols on gut microbiome composition and inflammation.

**Polyphenol Rich-Cereal(s)**	Dose	Population	Study Design	Duration	Gut Microbiome	Microbial Metabolites	Inflammatory Markers	Ref.
Oat bran and matched concentrations of β-glucan extract or polyphenol mix	1 and 3% *w*/*v*	3 female donors	In vitro batch fermentation	24 h	Increases in *Bifidobacterium*, and bacteria from the phyla of Proteobacteria and Bacteriodetes.	Increase in SCFA.	N/A	[101]
Black sorghum bran extract, sumac sorghum bran extract, fructooligosaccharides (FOS)	5 g/L	11 each of Obese and Healthy weight	In vitro batch fermentation	24 h	Sorghum bran polyphenols worked with FOS to enhance *Bifidobacterium* and *Lactobacillus*, and independently stimulated *Roseburia* and *Prevotella*.	No significant differences in total and individual SCFA production were observed between obese and healthy weight subjects.	N/A	[40]
Soluble feruloylated arabinoxylan oligosaccharides, rice bran polyphenols	50 mg	15 males and 17 females (21–45 years of age)	In vitro batch fermentation	24 h	Increased butyrogenic bacteria, *Coprococcus* and *Roseburia*.	No significant increase in SCFA.	N/A	[102]
Whole grain (WG) vs. refined grain (RG) foods	8 g (RG) 16 g (WG) per day	49 men, 32 women (ages 40–65 years)	Randomised, controlled, parallel-design human trial	6 weeks (after 2 weeks run-in Western-style diet)	WG group also showed more increase in stool weight, stool frequency and SCFA bacteria (*Lachnospira*) but decreased proinflammatory bacteria (*Enterobacteriaceae*).	N/A	No effect on salivary IgA concentration or stool IgA or stool cytokines concentrations. A higher percentage of terminal effector memory T cells and LPS-stimulated ex vivo production of TNF-α in WG than RG (*p* = 0.004).	[109]
Whole Grain Wheat (Whole vs. Refined)	70 g (WG) and 60 g (RW) per day	80 healthy overweight/obese subjects	A placebo-controlled, parallel-group randomised trial	8 weeks with measurements taken at baseline and every 4 weeks.	TNF-α reduction with WG consumption correlated with increased abundance of *Bacteroides* and *Lactobacillus* increases.	WG consumption increased serum dihydroferulic acid and faecal ferulic acid.	TNF-α and IL-10 reduced significantly	[60]
Whole grain (WG) versus Fruits and vegetables (FV)	3 servings/day	49 obese subjects	Randomised parallel-arm trial	6 weeks	No significant microbiota changes between groups were detected.	N/A	There was a significant decrease in LBP for participants on WG and FV diets with no change on the control diet. FV diet-induced significant change in IL-6 but no significant change in the other diets. WG diet resulted in a significant decrease in TNF-α whereas no significant effects by the other diets.	[108]
WG and RG	179 g/day (WG) and 13 g/day (RG)	60 Danish (20–65 year old with BMI 25–35 kg/m^2^)	A randomised, controlled crossover	Two 8-week intervention	Compared to RG, WG did not significantly induce major changes in faecal microbiome.	Plasma SCFA was not affected.	WG diet decreased bodyweight, serum inflammatory markers, IL-6 and CRP.	[107]
Purple wheat (PW) and regular wheat (RW)	4 servings/day (160 g total)	29 overweight and obese subjects	Randomised Single-blind parallel-arm study	8 weeks	N/A	No anthocyanins detected after 4 and 8 weeks. Ferulic acid and hippuric acid detected. No difference between PW and RW groups in terms of total phenolic acids.	IL-6 and adiponectin were reduced significantly with PW. TNF-α reduced in both groups. Plasma GSH level increased significantly in pooled data.	[110]
Black rice pigmented fraction (BFR) vs. white rice pigment fraction (WRF)	10 g	60 Coronary heart disease patients (45–75)	Randomised single-blind parallel-arm	6 months	N/A	N/A	No changes in plasma total superoxide dismutase activity and lipid levels. Reduced levels of vascular cell adhesion molecule-1(VCAM-1), high sensitive C-reactive protein (hs-CRP), soluble CD40 ligand.	[111]
Purple rice, red rice and brown rice	1 serving	22 obese participants	Crossover intervention trial	Over 4 h (30 min, 1 h, 2 h and 4 h time point)	N/A	N/A	A significant reduction in MDA levels with red rice and purple rice. IL-10 significantly reduced with purple rice (30 min). Red rice reduced IL-6 at 30 min and 1 h. Both purple and red reduced IL-12 at 10 min by 13.6% and 11.0% respectively. Brown rice did not show any effects on biomarkers.	[13]

N/A, Not Applicable; SCFA, Short-chain fatty acids; MDA, malondialdehyde; PW, purple wheat; RW, red wheat; WG, whole grain; RG, refined grain; FOS, fructo-oligosaccharide; VCAM-1, Vascular cell adhesion molecule 1; hs-CRP highly sensitive C reactive protein; LBP, liposaccharide binding protein; FV, Fruits and vegetables.

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
