# Peer review of "Bioaccessibility and Bioactivity of Cereal Polyphenols: A Review"

_foods, 2021, doi:10.3390/foods10071595_

Round 1

Reviewer 1 Report

General comments

The bioaccessibility and bioavailability of polyphenols have been reviewed by several authors recently. Notwithstanding, the subject is of actuality and reviews that target specific food matrix are very welcomed as the researches on phenolics bioaccessibility and bioavailability are ongoing and many aspects still to be investigated mainly in vivo.

This paper (review) provides for the first time specific insights on cereal polyphenols in a detailed, explicit and organized manner, using recent and trustful references. However, I can point out one weakness, where on many occasions the authors talk in a general manner about bioaccessibility and bioavailability or refer to other food products (tea, berry, wine, etc), which makes the content look similar to what has been published before (general reviews on polyphenols bioaccessibility and bioavailability). This can be accepted as there are not many studies on cereal phenolics. But this also decreases the specificity of the paper and can confuse the readers.

Specific comments

  • Introduction

The beginning of the last paragraph (line 51 to 59) does not seem to correlate with the structure of ideas.

(line 54 to 56) : Reference is messing

(line 57 to 64) : you should add stronger motivations.

In general the last paragraph should be rewritten. 

  • Main text

(line 72) 2. Cereal polyphénols:  In this section, the authors should talk about bound and free phenolics of the cereals matrix (as the binding of phenolics can be a limiting factor of their bioaccessibility).

Tbale 1: Columns titles do not reflect well the organization of the figures presented inside the table.

Reviewer 2 Report

The manuscript entitled “Bioaccessibility and Bioactivity of Cereal Polyphenols: A Review” needs to be revised . Some recommendations are as the following:

- The text contains several vague statements. For example, page 2, line 82, 88, 95: “strong antioxidant effects/activities” should be expressed quantitatively. Similarly, line 89: “increased remarkably”. How much was this increase? Or line 96-97: “low/high bioavailability”, again should be expressed quantitatively. Please read the entire text and instead of using vague statements such as “strong”, “high” or “low”, express these findings quantitatively. This issue is also present in the tables. For instance, table 2: “better recovery”. Instead of using subjective statements such as “better”, express the change in recovery as percentage.

- Another issue is related with the fact that the manuscript contains a lot of information on bioaccessibility and bioactivity of polyphenols from other sources besides cereals. For example, information on apple, propolis, tea and wine polyphenols are presented. I suggest authors to focus on cereal polyphenols.

- Page 6, line 162: Here I suggest authors to also refer to the following recent publication on the effect of food matrix on the bioavailability of polyphenols: doi: 10.1016/j.tifs.2020.10.030.

- Page 8, line 228: Here I suggest authors to also refer to the INFOGEST static in vitro gastrointestinal digestion method: doi: 10.1038/s41596-018-0119-1.

- Page 13: “and [73]”. Here please remove the “and”.

- Table 4: In this table please name the polyphenols that are responsible form the observed effects.

- Page 25, line 129: Nothing mentioned about gluten in the text until this point. This is an interesting aspect that should also be discussed in earlier, in the introduction perhaps.

Reviewer 3 Report

The review article is interesting. I have one suggestion that chapters 2.1.1 and 2.1.3 should be expanded with more reference examples.

Round 2

Reviewer 2 Report

The authors addressed majority of the points that are raised by the reviewer. The manuscript is now improved and therefore may be accepted for publication.